# Optimized Magnetization Distribution in Body-Centered Cubic Lattice-Structured Magnetoelastomer for High-Performance 3D Force–Tactile Sensors

**DOI:** 10.3390/s25072312

**Published:** 2025-04-05

**Authors:** Hongfei Hou, Ziyin Xiang, Chaonan Zhi, Haodong Hu, Xingyu Zhu, Baoru Bian, Yuanzhao Wu, Yiwei Liu, Xiaohui Yi, Jie Shang, Run-Wei Li

**Affiliations:** 1School of Materials Science and Chemical Engineering, Ningbo University, Ningbo 315211, China; houhongfei@nimte.ac.cn (H.H.); zhuxingyu@nimte.ac.cn (X.Z.); 2CAS Key Laboratory of Magnetic Materials and Devices, Ningbo Institute of Materials Technology and Engineering, Chinese Academy of Sciences, Ningbo 315201, China; xiangziyin@nimte.ac.cn (Z.X.); zhichaonan@nimte.ac.cn (C.Z.); huhaodong@nimte.ac.cn (H.H.); bianbr@nimte.ac.cn (B.B.); wuyz@nimte.ac.cn (Y.W.); liuyw@nimte.ac.cn (Y.L.); 3Zhejiang Province Key Laboratory of Magnetic Materials and Application Technology, Ningbo Institute of Materials Technology and Engineering, Chinese Academy of Sciences, Ningbo 315201, China; 4College of Materials Science and Opto-Electronic Technology, University of Chinese Academy of Sciences, Beijing 100049, China

**Keywords:** flexible tactile sensors, magnetoelastomer, lattice structure, magnetization direction, 3D printing

## Abstract

Flexible magnetic tactile sensors hold transformative potential in robotics and human–computer interactions by enabling precise force detection. However, existing sensors face challenges in balancing sensitivity, detection range, and structural adaptability for sensing force. This study proposed a pre-compressed magnetization method to address these limitations by amplifying the magnetoelastic effect through optimized magnetization direction distribution of the elastomer. A body-centered cubic lattice-structured magnetoelastomer featuring regular deformation under compression was fabricated via digital light processing (DLP) to validate this method. Finite element simulations and experimental analyses revealed that magnetizing the material under 60% compression strain optimized magnetization direction distribution, enhancing force–magnetic coupling. Integrating the magnetic elastomer with a hall sensor, the prepared tactile sensor demonstrated a low detection limit (1 mN), wide detection range (0.001–10 N), rapid response/recovery times (40 ms/50 ms), and durability (>1500 cycles). By using machine learning, the sensor enabled accurate 3D force prediction.

## 1. Introduction

Flexible devices exhibit high deformability, generating varying degrees of strain responses when bent, stretched, or compressed. By integrating flexible materials with functional materials through optimized structural design, the limitations of traditional rigid electronic devices are overcome [1,2,3]. Flexible tactile sensors, as a core component of flexible electronic devices, can provide multidimensional tactile perception for intelligent robots by imitating the mechanical response of human skin [4]. This helps to accurately detect the contact force [5], texture [6,7], and material properties [8,9] of the object, so as to achieve safe and dexterous interaction with complex environments for precise operation. And show revolutionary potential in the fields of medical health [10,11], human–computer interaction [12,13], and industrial manufacturing [14,15]. This is of great significance for improving the perception ability of the robot, improving the accuracy and adaptability of the operation, and expanding the application field.

At present, flexible tactile sensors with different sensing mechanisms have been widely studied, such as resistive [16,17], capacitive [18,19], piezoelectric [20,21], and magnetoelectric. Among these sensors, tactile sensing technology that uses magnetic sensors to detect changes in the amplitude and direction of the magnetic field has attracted much attention due to its three-dimensional force detection [22,23,24] and non-contact sensing capabilities [25].

The components of the flexible magnetic tactile sensor mainly include the soft mag-netic composite material as the source of the magnetic field signal, and the magnetic sen-sors based on the Hall effect [26], Giant Magnetic Resistance (GMR) [27,28], and Giant Magnetic Impedance (GMI) [29] effects as the sensing elements. These magnetic sensors can detect subtle changes in magnetic fields, enabling high-precision detection of stress signals. When the external pressure is applied, the shape or position of the magnetoelastic composite material changes, resulting in a change in the magnetic flux density around the magnetic sensor, and the external force information can be obtained.

The structural design and magnetization method of flexible magnetic material are important methods for improving the performance of magnetic tactile sensors. Based on the magnetic tactile sensor of flexible magnetic film and elastic support layer, researchers use different magnetization methods, such as folding magnetization [30], centripetal magnetization [31], and Halbach magnetization [32], to change the local magnetization direction of the magnetic film and increase the local magnetic properties of the magnetic film, enabling the detection of tangential force and improving the sensitivity. However, due to the high content of magnetic materials required for magnetic films and the large modulus, it is difficult to detect small contact forces (larger than 0.1 N). In order to enhance the detection limit of magnetic tactile sensors for small forces, based on the bionic structure of biological cilia, magnetic tactile sensors made of soft magnetoelastic materials with a cilia structure can obtain higher sensitivity and force resolution [33,34,35]. When an external force is applied, the cilia will bend significantly, resulting in a change in the direction of the magnetic field so that a small force can be detected. However, due to the low Young’s modulus of the cilia, the detectable force range is very limited (less than 30 mN). In addition, during the compression process, the bending of the cilia structure is irregular, resulting in inconsistent changes in the magnetic field and challenges in the force–magnetic coupling.

In this study, the challenges faced by magnetic tactile sensors in balancing sensitivity, detection range, and structural adaptability are addressed. We propose a pre-compressed magnetization method (the magnet is compressed before starting to magnetize) that can precisely control the magnetization distribution in body-centered cubic lattice-structured [36] magnetoelastomer (BLSM). As shown in Figure 1a, a BLSM was fabricated via digital light processing (DLP) to validate this method. The body-centered cubic lattice structure is selected because simulations demonstrate its regular deformation under compressive stress, with stress concentrations primarily occurring at the nodal junctions, while the pillars are rotated and without significant deformation, as shown in Figure 1b. This provides a controllable structure for precise regulation of magnetization distribution by the pre-compressed magnetization method. The effects of magnetization distribution and structural angles of BLSM on the magnetoelastic effect of BLSM are investigated. The experimental results show that BLSM can achieve a high magnetoelastic effect of 85%, which is mainly attributed to the 45° rotation of magnetization of the pillars from the horizontal to the normal direction. Finally, BLSM is integrated with a magnetic sensor into a flexible tactile sensor, demonstrating a low detection limit, high sensitivity, and wider detection range, alongside three-dimensional force prediction through machine learning. The sensor holds great potential for applications in intelligent robotics and human–computer interaction.

## 2. Materials and Methods

### 2.1. Materials

NdFeB magnetic particles (5 μm, New Nord, Guangzhou, China), coercive force (Hc) = 5.5 kOe, remanence (Br) = 8.91 kG. Flexible photosensitive resin Agilus30 (Stratasys, Eden Prairie, MN, USA), three-dimensional Hall sensor (MLX90393, Melexis, Glen, Belgium), and SiO_2_ (5 μm, Evonik, Darmstadt, Germany). Isopropanol (95%) was purchased from Shanghai Aladdin Biochemical Technology Co., Ltd. (Shanghai, China).

### 2.2. Manufacture of BLSM

The BLSM was prepared by digital light processing (DLP) 3D printing technology. The manufacturing process is shown in Figure 1a. Firstly, the photosensitive magnetic resin was prepared by mixing the solvent with NdFeB magnetic particles, Agilus30 resin, and SiO_2_. NdFeB is a high-performance magnet that can generate strong magnetic fields in confined spaces and maintain stable magnetic properties in complex environments due to its high coercivity, thereby resisting demagnetization [37]. The mass ratios of NdFeB, Agilus30 resin, and SiO_2_ in the magnetic composite resins with different contents are as follows: 10wt% (10wt% NdFeB: 87wt% Agilus30: 3wt% SiO_2_), 20wt% (20wt% NdFeB: 77wt% Agilus30: 3wt% SiO_2_), 30wt% (30wt% NdFeB: 68wt% Agilus30: 2wt% SiO_2_), and 40wt% (40wt% NdFeB: 58wt% Agilus30: 2wt% SiO_2_). Next, a planetary mixer is used for multistage operation to fully mix the magnetic particles in the resin. In the third step, the designed 3D model file is sliced into 25 μm layers and imported into the DLP 3D printer (AUTOCERA-L, Beijing Shiwei Technology Co., Ltd., Beijing, China) to prepare the BLSM. The printed magnetoelastic material is removed from the printing platform, immersed in an isopropanol solution, ultrasonically cleaned for 5 min (25 kHz), and then dried in a well-ventilated area. Finally, a custom fixture is used to magnetize the magnetoelastic material under a certain compressive deformation under a strong magnetic field of 2 T. After magnetization, the fixture is removed, and the material changes the direction of the internal magnetic pole due to its own rebound.

### 2.3. Preparation of the Magnetic Tactile Sensor

The magnetic tactile sensor is composed of upper, middle, and lower layers, as shown in Figure 1a. The upper layer is a BSTM, the middle layer is a hard polymer for isolating magnetoelastic materials and magnetic sensors, and the lower layer is a flexible printed circuit board (FPC) with a three-dimensional magnetic sensor (MLX90393). The BLSM was secured to the bottom isolation layer using double-sided adhesive tape. This fixing method does not affect the deformation of BLSM due to its low modulus. The flexible circuit board is connected to the single-chip microcomputer (Arduino UNO R3, Monza, Italy). The single-chip microcomputer is connected to the PC through USB, and the serial port debugging software SSCOM 5.13.1 is used to display and store the three-dimensional magnetic field size data on the PC.

### 2.4. Characterization

Magnetic composite materials were manufactured using a 3D printer, and the stress–strain curve was obtained through tensile testing performed with a universal testing machine (Instron 5943, Norwood, MA, USA). The magnetic properties of magnetic composite materials with different magnetic particle contents were characterized by a vibrating sample magnetometer (Lake Shore, VSM-7410, Westerville, OH, USA). The universal testing machine (Instron 5943, Norwood, MA, USA) characterizes the force–magnetic properties of the prepared magnetic tactile sensor. The indenter moves along the Z-axis to apply normal stress to deform the magnetic elastomer. The three-axis magnetic sensor (MLX90393) is used to collect space magnetic signals. The microcontroller is used to control the three-axis Hall sensor and read real-time data to evaluate its force–magnetic conversion performance.

### 2.5. Computational Simulations

A finite element simulation was performed using COMSOL Multiphysics 6.1. The solid mechanical components are used to simulate the deformation and stress distribution of the magnetic elastomer. The bottom of the magnetic elastomer is set to a fixed constraint, and the rest is set to free. A boundary load is added at the top to cause compression deformation. The deformed geometric model can be obtained for subsequent magnetic simulation. Subsequently, the component (magnetic field, no current) is used to simulate the magnetic field distribution after the compression deformation of the magnetic elastomer, and the magnetic field distribution is simulated by setting the residual magnetic flux direction of the magnetic elastomer.

## 3. Results

### 3.1. Magnetic and Mechanical Properties of Magnetic Composite Material

The mass ratio of magnetic particles to flexible light-cured resin significantly affects the magnetic and mechanical properties of magnetic composite materials. The magnetic and mechanical properties of magnetic composite materials with different magnetic particle contents were characterized. The results are shown in Figure 2a,b. With the increase in the mass ratio of magnetic particles, the magnetic properties of magnetic composite materials are enhanced, and a higher content of magnetic particles is beneficial for improving the force–magnetic response. Magnetic composites with different particle contents have the same coercivity and no chemical interaction. However, when the content of magnetic particles increases, the Young’s modulus of the magnetic composite material increases, and its flexibility decreases. Increasing the content of magnetic particles can enhance the change in magnetic flux density during the compressive deformation of magnetic composite materials. When the content of magnetic particles reaches 40 wt%, the particles tend to aggregate, resulting in lower uniformity of the mixture (see Appendix A). Uneven distribution of magnetic particles can cause an uneven distribution of stress within the material, leading to a concentration of stress in dense areas, which become the weak points of the material. The scanning electron microscope (SEM) images and the elemental distribution maps of the cross sections of the 30 wt% samples were analyzed and observed (see Appendix A), and it was found that the magnetic particles were uniformly distributed in the matrix material. For these reasons, a mass ratio of 30 wt% was ultimately selected for the preparation of BLSM. The BLSM sample is shown in Figure 2c, which has high printing accuracy. The size of the bcc unit is 2.5 mm × 2.5 mm × 2.5 mm, and the diameter of the pillar is 0.6 mm. The compression test of BLSM is shown in Figure 2d. It is worth noting that BLSM has high compressibility, can produce huge deformation in the low-stress range, and can withstand large compressive stress.

### 3.2. The Effect of Magnetization on the Force–Magnetic Conversion Performance of BLSM in Compression State

The magnetization distribution of magnetoelastic materials can be effectively modulated through magnetizing the magnetoelastomer under different compressive strains. A series of BLSM with the same structural parameters (a = b = c = 2.5 mm, α = 70.6°) were manufactured (see Appendix A) and magnetized under different compression states of 0% (C-0), 20% (C-20), 40% (C-40), and 60% (C-60), respectively, as shown in Figure 3a. After the applied stress was released, these magnetoelastomers demonstrated distinct magnetization distributions. The magnetization direction of the BLSM pillars was analyzed in the initial state, and the result is shown in Figure 3b. Since the BLSM unit contains eight identical pillars, which are symmetrically distributed around the body center node, one pillar was selected to analyze the variation in the magnetization direction for simplicity. According to the mechanical simulation of the BLSM compression process (Figure 1b), it is seen that the upper and lower pillars are rotated around the center nodes, which indicates that the magnetization direction of the pillars changes with the rotation of the pillars. In order to quantitatively describe the magnetization distribution inside the BLSM, θ is defined as the angle between the direction of the magnetization and the horizontal plane.

As shown in Figure 3b, through precise control of compression, the initial magnetization directions of the pillars in the 0%, 20%, 40%, and 60% pre-compressed states were 90.0°, 103.6°, 115.1°, and 125.3°, respectively. As the angle of the magnetic field direction deviates from the vertical direction (90.0°), the initial magnetic flux density gradually decreases. This result indicates that intentionally adjusting the magnetization distribution to the horizontal direction can reduce its magnetic flux density. A small initial magnetic flux density is required, suggesting that this magnetization distribution (125.3°) is beneficial for achieving a large magnetoelastic effect, as determined by the ratio ∆B/B_0_ (where B_0_ is the initial magnetic flux density without external deformation). The magnetization direction of the pillars in the above BLSMs with a 60% compressive state was analyzed as well, and the result is shown in Figure 3c. It is shown that for the magnetization directions of pillars in the 0%, 20%, 40%, and 60% pre-compressed states, magnetization direction angles of 54.7°, 68.3°, 79.8°, and 90.0° are observed, respectively. It is found that as the direction approaches the vertical (90.0°), the detected magnetic flux density increases, as shown in Figure 3d,e. For the elastomer pre-compressed at the 60% state and magnetized, the relative change in magnetic flux density reached up to 55.85%. Additionally, a magnetization rotation of 35.3° from the horizontal to the vertical direction was observed, which was the largest rotation among all the samples.

### 3.3. The Influence of BLSM Structure Parameters on the Force–Magnetic Conversion Performance

The magnetization direction of BLSM can also be effectively controlled through the angle configuration of the BCC lattice structure. In the previous section, we found that the BLSM with a structure angle of 70.6° was magnetized at 60% compressive deformation, allowing a 35.3° (Δθ) rotation of the magnetization direction to achieve a higher change in magnetic flux density. We selected BLSMs with structural angles of 60° (Δθ < 35.3°) and 90° (Δθ > 35.3°) for the follow-up study to demonstrate the effect of the rotation angle of the magnetization direction on the flux density variation.

We systematically defined the BLSM unit parameters (see Appendix A) and fabricated a series of BLSMs with structural angles of 60.0°, 70.6°, and 90.0°. These BLSMs had the same thickness and pillar diameter (2.5 mm and 0.6 mm, respectively). Under a 60% compression state, the magnetization direction of the BLSM pillars in all three samples is 90°. When the samples return to their original state, the magnetization directions within the elastomer rotate to 120.0°, 125.3°and 135.0°, respectively, as shown in Figure 4a. The angular variation in the magnetization direction during the compression strain process (from 0% to 60%) is shown in Figure 4b. The results show that during compression, the magnetization direction within the magnetoelastic material gradually rotates toward the vertical direction (90°). The variation in magnetic flux density during the compression strain process is shown in Figure 4c. It is noteworthy that the elastomer with a 90° structural configuration undergoes a 45.0° directional rotation during compression, thus exhibiting the greatest change in magnetic flux density. In contrast, the elastomer with a 60.0° structural configuration undergoes a 30.0° rotation during compression, and its magnetic flux density change is smaller compared to that of the 70.6° sample. The relative magnetic flux density change in BLSM (90°) is as high as 85%. This significant change indicates that the alteration of the magnetization direction affects the magnetic flux density of the elastomer, and the magnetic flux density can be adjusted through the compression state and the structural state of the elastomer.

### 3.4. Mechanism Analysis

According to the above research, the change in magnetic flux density increases during the compression process of BLSM. This phenomenon is due to two factors: (1) the distance between the magnetic particles within the magnetoelastic material and specific locations gradually decreases; (2) the magnetization direction within the elastomer rotates. As shown in Figure 5a, to clarify the contribution of magnetization direction rotation to the change in magnetic flux density, we systematically studied the magnetic flux density generated by a single magnetic unit at the target position under different magnetization direction configurations. A magnetic sphere with a programmable magnetization angle was developed using COMSOL Multiphysics 6.1 software. A reference point was selected (1 mm below the sphere) to measure the vertical component of the magnetic flux density (Bz) under different magnetization configurations. The results, as shown in Figure 5b, indicate that as the magnetization direction rotates from parallel (0° or 180°) to vertical (90°), Bz increases monotonically. This trend indicates that the rotation of the magnetization direction toward the vertical significantly enhances the magnetic flux density in that direction.

To clarify the evolution mechanism of magnetic flux density in BLSM under compression, we conducted additional simulations in COMSOL, focusing on the evolution of magnetization direction along the diagonal of the BLSM, as shown in Figure 5d. Based on the structural deformation states of BLSM at different compressions, the deflection of the magnetic pole angle is primarily due to the rotation of the pillars. Since there is no obvious rotation at the pillar connection, we set the boundary condition of each pillar in the BLSM with a change in magnetization direction, while the magnetization direction at the connection remains unchanged (see Appendix A). The diagonal vector of the BLSM is in the <111> direction, and the magnetization directions of the pillars were calculated and set (see Appendix A). As shown in Figure 5c, it is noteworthy that the evolution of the simulated magnetic flux density under compression strain is consistent with the experimental measurement results, which jointly reveal the force–magnetoelastic coupling mechanism of BLSM. Specifically, changes in the spatial distribution of magnetic particles [38] and the direction of magnetization [39] under mechanical deformation combine to produce changes in BLSM magnetic flux density.

## 4. Application

When magnetoelastic materials are subjected to mechanical stress, changes in the magnetic field occur, which can be combined with magnetic sensors to achieve tactile sensing functions. Therefore, we used BLSM as the magneto-sensitive material and Hall sensors as the magnetic sensing units for flexible tactile sensors (Appendix A). The thickness of the magnetoelastic material is 5 mm, which is comparable to the values reported in previous studies [31,32,35].

The response value of the magnetic tactile sensor is defined as ΔB (i.e., B-B_0_), which refers to the relative change in magnetic flux density between B after the application of force and B_0_ under no-load conditions. The sensitivity of the magnetic tactile sensor is defined as ΔB/ΔF, where F represents the relative change in applied pressure within the linear response range. As shown in Figure 6a, to characterize the sensor’s response to normal pressure, pressure was applied in the range of 0–10 N, and the change in magnetic flux density was examined. The sensor’s sensitivity at the initial, middle, and final stages is 1688.5 μT/N (0–0.16 N), 225.8 μT/N (0.16–3.56 N), and 47.5 μT/N (3.56–10 N), respectively. The initial segment rises quickly, while the later segment rises slowly. The rapid rise in the initial segment is due to the increase in normal pressure, which causes deformation that not only brings the magnetoelastic material closer to the magnetic sensor but also causes the BLSM lattice structure to rotate, resulting in a rotation of the magnetic field direction toward the vertical, leading to a change in magnetic flux density. The slow rise in the later segment is due to the limitations of the rotation of the magnetoelastic lattice and the change in distance, which causes the variation in magnetic flux density to approach saturation, resulting in a smaller change in magnetic flux density.

Additionally, the performance characteristics of flexible sensors, such as stress detection limits, dynamic response time, and robustness, are also important indicators for evaluating the sensor. In order to measure the detection limit and resolution of the sensor for small forces, pressures of 0.001 N, 0.002 N, 0.003 N, and 0.004 N were applied to the sensor, and each pressure was held for 5 s, as shown in Figure 6b. As these pressures were applied sequentially, the magnetic flux density gradually increased. The fluctuation in the magnetic signal variation curve at 0 N pressure in Figure 6b is the noise of the sensor, which is mainly influenced by the ambient magnetic field during the testing process.

The sensor response time is the time from the application of an external force to the stability of the output signal, while the recovery time is the time from the release of the force to the return of the signal to its initial state. To evaluate the response time of the sensor, rapid compression and recovery tests were performed on the elastomer, using an Instron testing machine to apply a 0.3 mm instantaneous displacement (0.015 N) at a loading and unloading speed of 500 mm/min to obtain the dynamic response time of the sensor. As shown in Figure 6c, the response time and recovery time are 40 ms and 50 ms, respectively, indicating that the sensor exhibits good response and recovery speed. To detect the dynamic force response characteristics of the sensor, the sensor was subjected to dynamic pressure inputs of 0.35 N, 0.75 N, 1.50 N, and 2.00 N (10 loading and unloading cycles), and the corresponding changes in magnetic flux density were measured. The results shown in Figure 6d indicate that the sensor’s magnetic flux density increases with pressure, demonstrating that the sensor has reliable, stable, and repeatable sensing behavior. For flexible tactile sensors, robustness is a very important performance characteristic. The sensor was tested with 1500 load/unload cycles at a compression displacement of 1 mm (0.4 N). The results, as shown in Figure 6e, indicate that the sensor output shows no significant drift as the number of tests increases, indicating good repeatability and durability.

Finally, since the magnetic sensor used can detect the magnetic flux density output in three directions, a force with a shear component is applied to the 3D force sensor to test its 3D force sensing capability. The results are shown in Figure 6f. When the X-direction shear force increases to 4.5 N, the magnetic flux density reaches saturation. The magnetic response of the Y-direction shear force can be found in Appendix A by recording the applied force (Fx, Fy, Fz) and the corresponding Hall effect sensor output (Bx, By, Bz). The Back Propagation (BP) neural network predicts the 3D force by learning the mapping relationship between the magnetic signal and the force. Data on 3D forces and corresponding magnetic signals from magnetic tactile sensors under different forces are collected and used as training samples. After sufficient training, the network is able to capture the complex relationship between the magnetic signals and the forces. It can accurately predict the corresponding 3D force when a new magnetic signal is input. We collected 500 data sets, of which 450 groups were used for neural network training, and another 50 data sets were used to verify the model’s prediction of force in different directions. The results are shown in Figure 6g. RMSPE (Root Mean Squared Percentage Error) is a metric for evaluating the error of predictive models, particularly in regression problems. It measures the performance of a model by calculating the square root of the average of the squared relative errors between the predicted and actual values. The formula for calculating the RMSPE is:(1)$$\mathrm{RMSPE}=\sqrt{\frac{1}{n}\sum_{i=1}^{n}{\left(\frac{y_{i}-y}{y_{i}}\right)}^{2}}$$

In this formula, *y_i_* is the experimental value, *y* is the predicted value, and *n* is the number of observations. According to the calculation, the prediction errors of the model for F_X_, F_Y_, and F_Z_ are 2.04%, 2.7%, and 3.64%, respectively, and the accuracy of the model in predicting forces is 97.96%, 97.3%, and 96.36%, respectively. The model has the ability to detect the size of the tangential force and identify the direction of the force. Subsequent research will expand the training data set to optimize the training model.

By comparing the sensitivity, resolution, measurement range, and three-dimensional force detection abilities of magnetic tactile sensors in recent years, as shown in Table 1, we find that our magnetic tactile sensor has an advantage in force resolution. It also has a wide detection range and can be used to detect the magnitude and direction of three-dimensional forces. Therefore, it is more suitable for robotic hands to grasp and detect different objects.

## 5. Conclusions

This study proposes a method of magnetizing the body-centered cubic lattice-structured magnetoelastomer (BLSM) under a compressed state to fabricate magnetoelastic materials with a high magnetoelastic effect. Finite element simulations and experimental analyses revealed that magnetizing the magnetoelastomer under 60% compression strain gives rise to the optimized magnetization distribution, enhancing force–magnetic coupling by enabling a maximum 45° rotation in the magnetization direction. The relative change in magnetic flux density is 85%. The BLSM and the magnetic Hall sensor form a magnetic tactile sensor, which demonstrates excellent performance, including a low detection limit (1 mN), wide detection range (0.001 N to 10 N), fast response/recovery time (40 ms/50 ms), excellent durability (>1500 cycles), and high stability. By using neural network algorithms, sensors can predict 3D forces, with a maximum error percentage of 3.56% (Accuracy rate 94.44%), highlighting their potential in complex tactile sensing. Future research will continue to focus on the design of magnetoelastomer structures to optimize the distribution of magnetization directions for improved magnetic tactile sensors for high-sensitivity and high-precision operation.

## Figures and Tables

**Figure 1 sensors-25-02312-f001:**
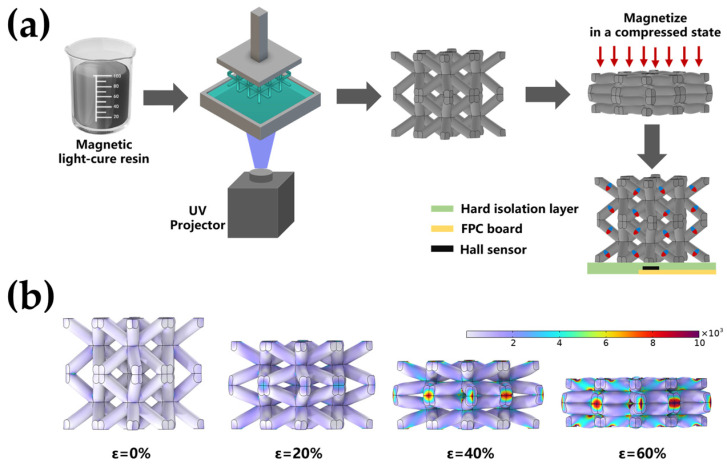
(**a**) Fabrication of BLSM, magnetization method, and flexible tactile sensor. (**b**) Mechanical simulation of BLSM.

**Figure 2 sensors-25-02312-f002:**
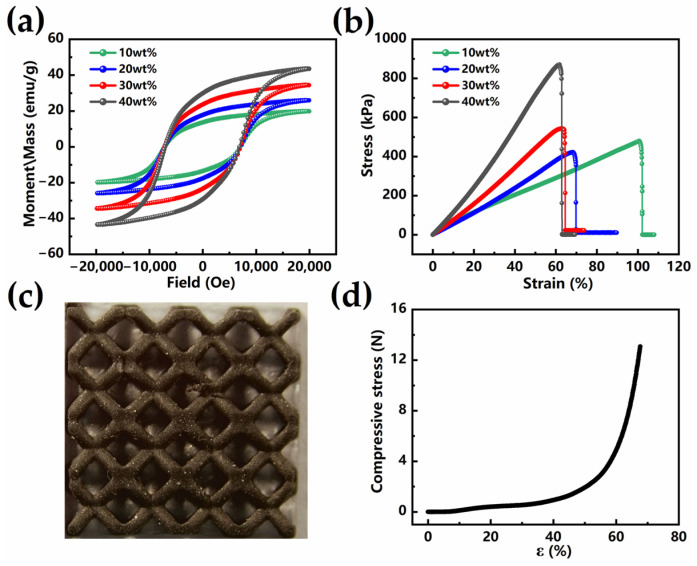
(**a**) VSM test results of magnetic composite materials with different magnetic particle contents. (**b**) Stress–strain curves of magnetic composite materials with different magnetic particle contents. (**c**) BLSM top view. (**d**) Compressive stress—compression curve for the BLSM.

**Figure 3 sensors-25-02312-f003:**
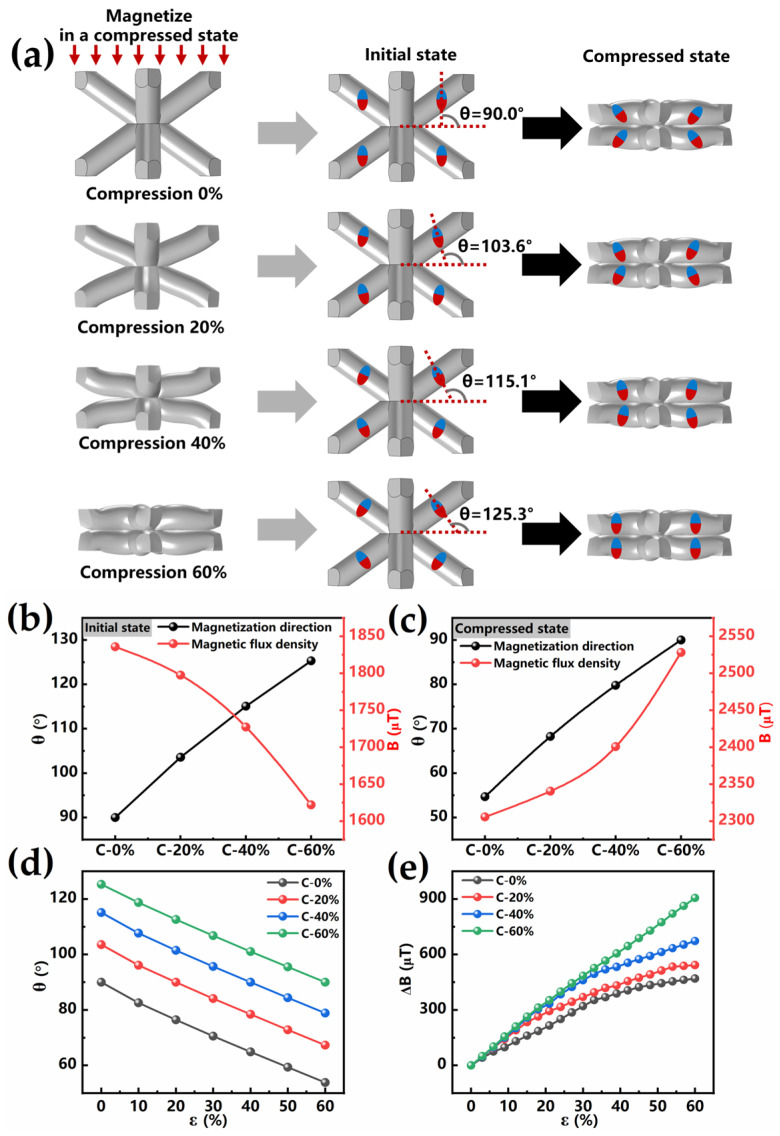
Magnetization direction and magnetic flux density of magnetization in different compression states: (**a**) Magnetization direction of initial and compressed (C-60%) state. Inset: micromagnets (red and blue spheres) in the matrix material. (**b**) Initial magnetization direction angle and magnetic flux density. (**c**) Magnetization direction angle and magnetic flux density in compressed state (C-60%). (**d**) The variation in the magnetization direction angle with compression. (**e**) The variation in the magnetic flux density with compression.

**Figure 4 sensors-25-02312-f004:**
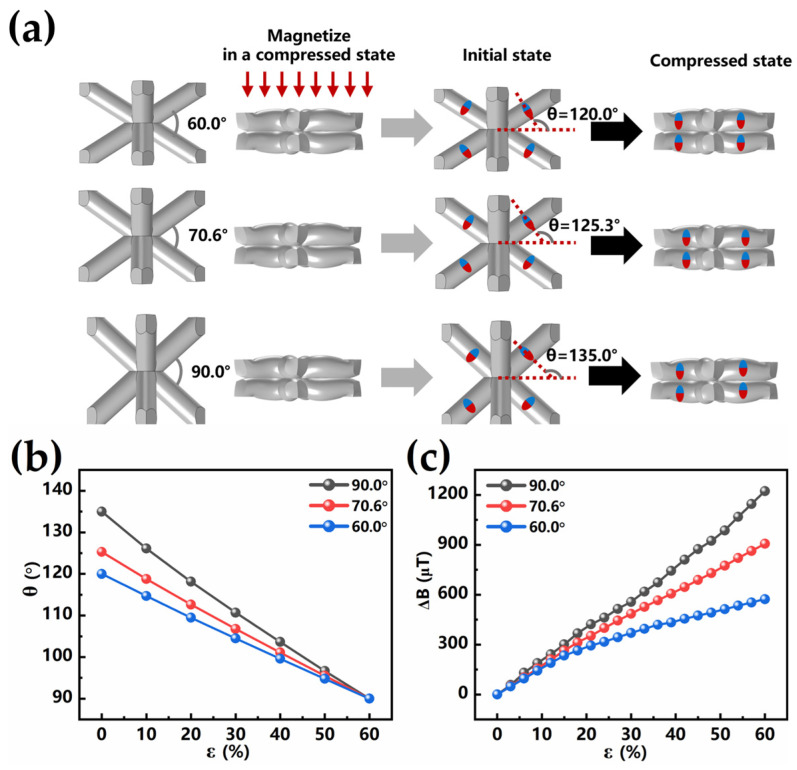
Magnetization direction distribution and magnetic flux density in different structural angles: (**a**) Magnetization direction of initial and compression (C-60%) state. Inset: micromagnets (red and blue spheres) in the matrix material. (**b**) The variation in the magnetization direction angle with compression. (**c**) The variation in the magnetic flux density with compression.

**Figure 5 sensors-25-02312-f005:**
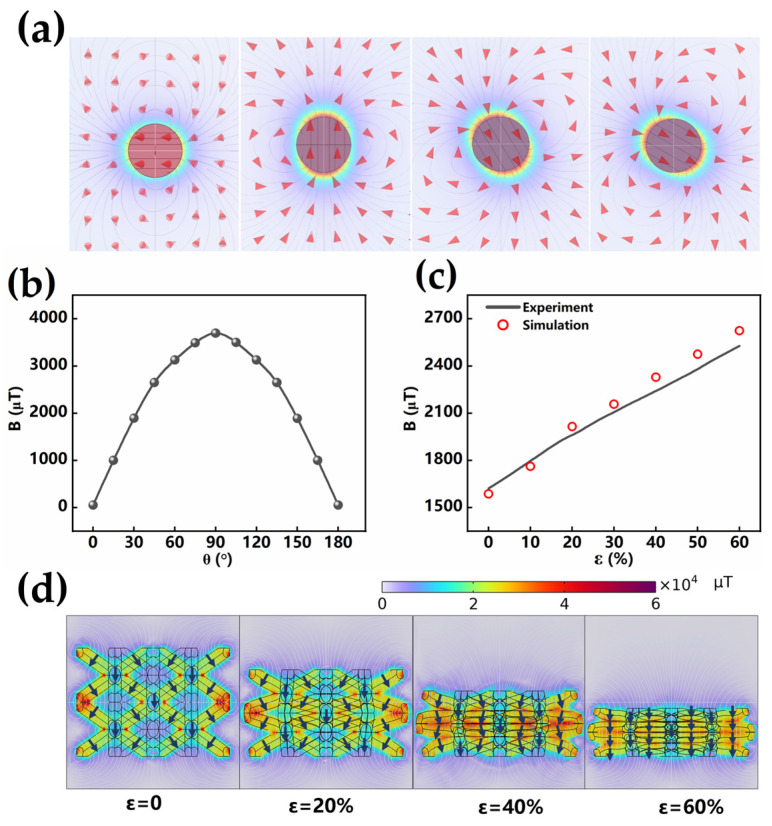
Magnetic performance simulation. (**a**) Different magnetization directions of magnetic spheres. (**b**) Magnetic flux density in different magnetization directions. (**c**) Magnetic flux density in simulation and experiment (**d**) Magnetization model in different compression states.

**Figure 6 sensors-25-02312-f006:**
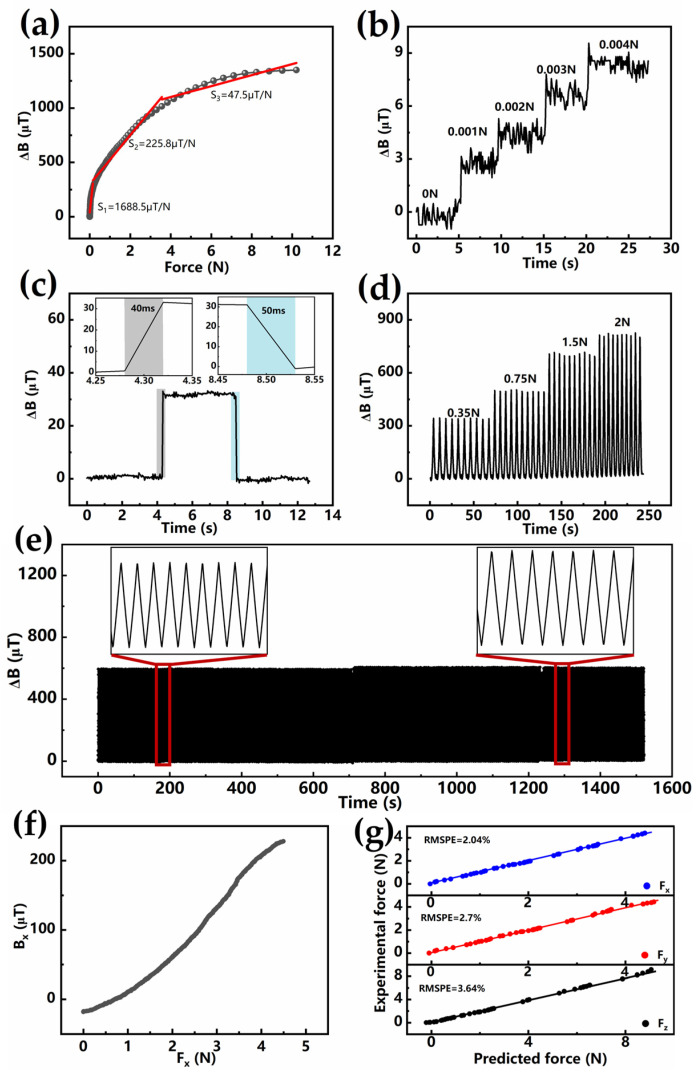
Performance results of the sensor. (**a**) Response of the sensor to pressure. (**b**) The detection limit of the sensor. (**c**) Dynamic response time and Magnetic response under different pressures. (**d**) Repeatedly compressed sensor output under different pressures. (**e**) Durability test results of more than 1500 cycles. (**f**) The magnetic response of the tangential force parallel to the X-direction. (**g**) The relationship between the prediction force and the experimental force.

**Table 1 sensors-25-02312-t001:** Comparison of different magnetic tactile sensors.

Ref.	Sensitivity	Resolution	Range	3D-Force Sensing
[23]	/	30 mN	1.9 N	Only direction
[24]	16 mV/N	100 mN	20 N	Yes
[40]	/	1.5 N	10 N	No
[32]	/	10 mN	10 N	Yes
[35]	6.63 μT/mN	0.2 mN	19.5 mN	Only direction
This work	1688.5 μT/N	1 mN	10 N	Yes

“/” indicates that there is no relevant information in the paper.

## Data Availability

The original contributions presented in this study are included in the article/Appendix A. Further inquiries can be directed to the corresponding authors.

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
