# Peer review of "Optimized Magnetization Distribution in Body-Centered Cubic Lattice-Structured Magnetoelastomer for High-Performance 3D Force–Tactile Sensors"

_sensors, 2025, doi:10.3390/s25072312_

Round 1
Reviewer 1 Report
Comments and Suggestions for Authors
The reviewed paper is dedicated to the development of flexible magnetic tactile sensors based on magnetoelastic materials with a body-centered cubic (BCC) lattice structure fabricated via digital light processing (DLP). The authors propose an innovative pre-compressed magnetization strategy to enhance the magnetoelastic effect, demonstrating that the optimal magnetization orientation is achieved when the material is pre-compressed by 60%. This solution enabled a significant increase in magnetoelastic effect, reaching up to 85% relative change in magnetic flux density.
The experimental results show a low detection limit of 1 mN, a broad sensing range (0.001–10 N), and impressive dynamic performance, including response and recovery times of 40 ms and 50 ms respectively. The sensor also demonstrated excellent durability, verified through cyclic tests exceeding 1500 cycles.
However, it would be beneficial to discuss more extensively the potential limitations of the proposed structure when scaled and the effects of varying external operating conditions (e.g., temperature and humidity).
The text needs correction of a number of typos and unclear expressions.
Line 65. "...based on the bionics of biological cilia," - not clear.
Line 75-76. The word "structure" is lost.
Line 81. Change "magnetization arrangement" to "magnetization distribution".
Line 125-126. The sentence "The stress-strain curve was obtained by using 3D printer..." seems to be a mistake.
Line 150-152. Very strange curing that "The magnetic properties and mechanical properties of standard parts with different magnetic particle contents were characterized by photocuring printing. The results are shown in Figure 2a." In this figure the VSM curves are presented. So the statement must be corrected.
Line 155. Change "When" to "when".
Line 178. Change "magnetization arrangement" to "magnetization distribution".
Line 185. Change "the pillars with the rotation of pillars" to "...the pillars changes with the rotation of pillars".
Line 195. Change "...large magnetic-elastic effect..." to "...large magnetoelastic effect...".
Line 208. Change "compression" to "compressed".
Figure 6 a. Correct "Froce" to "Force".
Line 347. Correct "magneto elastomer magnetoelastic under" to "magnetoelastomer under".
Line 349-350. Change "The relative change of the magnetic flux density upon 85% is around." to "The relative change in magnetic flux density is approximately 85%".
Comments on the Quality of English Language
the text of the article needs careful reading and correction of stylistic inaccuracies and typos
Reviewer 2 Report
Comments and Suggestions for Authors
In this manuscript (sensors-3533730) reported a body-centered Cu-bic-structured magnetoelastomer for high-performance 3D force-tactile sensors. The prepared force tactile sensor performance results are acceptable, but it still lacks innovative material design. In addition, there are many issues with experimental details, writing, and discussion. The manuscript needs a major revision before possible acceptance.
1. The title mentions the body-centered Cu-bic-structure, but the introduction does not explain the advantages of choosing this structure. Additionally, regarding the choice of magnetic materials, there is no explanation provided for why NdFeB is used.
2. The title mentions a high-performance 3D force-tactile sensor. Please provide a table to compare the results of this paper with those in previous
3. 102 line: The mass ratio of each material in the mixed solution needs to be provided with clear data.
4. How are magnetic elastic materials fixed on the hard isolation layer, and will this fixation affect the strain properties of magnetic elastic bodies?
5. The selection of a magnetic inclusion of 30 wt% based on Figures 2a and b is not convincing enough, since it seems acceptable to obtain a result of 20 wt% from Fig. 2.
6. Why does the microstructure of the sample cross-section affect the mechanical properties of magnetoelastomer?
7. 216 lines: What is the basis for the selection of the three values of BSM structure angles: 60.7°, 70.6°, and 90.0°?
8. 268 lines: The reduction in the distance between magnetic particles leads to the conclusion of this paper. Please provide relevant evidence and references.
9. Definition of response/recovery time, response value and sensitivity of a magnetic sensor.
10. 286 line: A linear fitting curve for the sensitivity of the sectional curve should be provided.
11. How many Newtons is the force applied to the figure?
12. What method of the neural network algorithm used in this paper needs to be explained. According to practical applications, the final value predicted by the mechanical sensor should represent the resultant force in the X, Y, and Z directions. What is the accuracy of the predicted force and the experimental force?
Reviewer 3 Report
Comments and Suggestions for Authors
The authors investigate the feasibility of a flexible magnetic tactile sensor made of a mixture of elastomer and NdFeB. The study is methodologically correct and the results seem interesting. However, the following issues need to be addressed before the manuscript is considered for publication:
- The use of the terms: “body-centered cubic-structured magnetoelastomer” or “body-centered cubic lattice” in the title and keywords, respectively, but also throughout the text, is confusing. Usually, these terms refer to the arrangement of atoms in a material, therefore one might think that the atoms of the polymer arranged in a BCC structure, which is not the case. For this reason, the authors should find another expression, rather such as “cubic diagonal framework”, to accurately reflect their case.
- In the supplementary material the authors show an SEM image of one of the samples as evidence of the uniform dispersion of NdFeB in the polymer matrix. However, this is inconclusive and an elemental mapping (perhaps only for Nd, Fe, and B) or backscattering images might be more informative.
- Line 102: “The mixture was mixed according to the mass ratio” is not only unclear but the authors should clearly write these ratios in this section.
- At the beginning of the subsections 3 and 4 the authors should remove the explanatory text from the original form.
- Line 283: “which is comparable to the values reported in previous studies” these studies should at least be cited.
- The conclusions section should be rewritten to clearly summarize the results. For example, statements such as “The relative change of the magnetic flux density upon 85% is around.” should be avoided.
- There are many annoying repetitions throughout the text that should be corrected. Some examples: line 128 “The tensile test tensile machine”, line 297 “detecting the detection limits”, line 346 “magnetizing the magneto elastomer magnetoelastic”.
- This sentence is ambiguous and should be corrected: This trend indicates the rotation of the magnetization direction from directly give rise to the enhancement of magnetic flux density.
The language should be carefully revised. There are hastily written sentences and annoying repetitions. Examples are given in the comments section.
Reviewer 4 Report
Comments and Suggestions for Authors
Submitted manuscript “Optimized Magnetization Arrangement in Body-Centered Cubic-Structured Magnetoelastomer for High Performance 3D 3 Force-Tactile Sensors” describes valuable set of interesting experiments and modelling related to the development of particular type of the force-tactile sensors. This technological direction and corresponding scientific basis are not exactly new and for decade it was addressed as flexible sensors or even artificial skins (Melzer, M., Makarov, D., Calvimontes, A., Karnaushenko, D., Baunack, S., Kaltofen, R., Mei, Y., et al. Stretchable magnetoelectronics (2011) Nano Letters, 11 (6), pp. 2522-2526. doi: 10.1021/nl201108b; Ferńandez, E., Kurlyandskaya, G.V., García-Arribas, A., Svalov, A.V. Nanostructured giant magneto-impedance multilayers deposited onto flexible substrates for low pressure sensing (2012) Nanoscale Research Letters, 7, art. no. 230, pp. 1-5 doi: 10.1186/1556-276X-7-230; and others). Proper analysis or the concepts must be given in the Introduction accompanied by citations related to main directions and contributions. This comparative overview will benefit the work which is poorly referenced in the sense of reflection of the important international contributions.
The second shortcoming comes from the attempt to add additional significance to the described research and use too many new/invented definitions. Abbreviation bcc in the solid state physics is reserved for body cubic centered crystalline structure. Authors may use it (if they would not like to avoid this confusion), however just in inverted commas - “bcc”. Another problem is the usage of ornately florid language which is not appropriate for research publication and simply non accurate. For example, what does it mean “pre-compressed magnetization strategy”? Magnetization is either magnetic moment per unit of mass/volume or the process of the change of M value under application of the external magnetic field (sometimes it can be used for the change of M under temperature variations). Try to avoid word “strategy” – simple technological solutions are not strategy, at least they should not be defined like this by the authors of publication. The same case for “synergistic optimization of BCC structure parameters and magnetization arrangement through the precompression magnetization”. The concept of the shape anisotropy does not even appear. Why? If the magnetoelastic one is dominating it should be discussed.
Any research work must have the goal of the research given as a research/technological problem as opposed to the research plan or simple description of the steps taken in the course of investigation. Formulate the research problem at the end of the Introduction as 3-5 lines paragraph.
The study has severe shortcoming in the part of magnetic measurements and results interpretation. There are no structural, chemical and magnetic characterization of the magnetic filler and the matrix themselves as well as M(H) data for simple shape cylindrical composite. Include the interaction part analysis (Safronov, A.P., Mikhnevich, E.A. Magnetostriction in ferrogels based on physical and chemical networking with embedded strontium hexaferrite particles Journal of Physics: Conference Series , 2019, 1389(1), 012057 DOI 10.1088/1742-6596/1389/1/012057; Stolbov, O.V., Raikher, Y.L., Balasoiu, M. Modelling of magnetodipolar striction in soft magnetic elastomers (2011) Soft Matter, 7 (18), pp. 8484-8487. doi: 10.1039/c1sm05714f). Ensemble of the particles should have the particle size distribution?
Figures must show error bars or experimental errors should be discussed additionally in the main text.
Comments on the Quality of English LanguageAnother problem is the usage of ornately florid language which is not appropriate for research publication and simply non accurate. For example, what does it mean “pre-compressed magnetization strategy”? Magnetization is either magnetic moment per unit of mass/volume or the process of the change of M value under application of the external magnetic field (sometimes it can be used for the change of M under temperature variations). Try to avoid word “strategy” – simple technological solutions are not strategy, at least they should not be defined like this by the authors of publication. The same case for “synergistic optimization of BCC structure parameters and magnetization arrangement through the precompression magnetization”.
Round 2
Reviewer 2 Report
Comments and Suggestions for Authors
The manuscript has been revised accordingly and could be accepted in present form.
Author Response
We are very grateful for your careful review. We have examined and verified this manuscript carefully, and we also have worked on both language and readability and made every effort to address any potential issues in our manuscript. Your professional feedback has provided significant help in improving the quality of the research.
Reviewer 3 Report
Comments and Suggestions for Authors
The authors have improved the manuscript and it is now ready for publication.
Author Response

(The authors gave the same response as above.)

Reviewer 4 Report
Comments and Suggestions for Authors
Submitted revised version of the manuscript shows some improvements. Authors made an effort to answer the questions. However, there are statements which are simply wrong or just non-objective. Authors placed an opinion ( lines 55-57) “However, the limitation of their detection range and the difficulty of detecting 3D magnetic field changes have restricted their wide application in the field of tactile sensing”. I am very sorry to say that without clear definition of the parameter for the detection range this statement is simply not valid. Manuscript still has very narrow citation range for existing fundamental contributions (https://doi.org/10.1007/s10853-015-9643-3; https://doi.org/10.1038/ncomms4266; https://doi.org/10.1088/0957-4484/22/36/365304, etc.)
In addition, proposed solution has strong limitation in sense of the size and temperature range of functionality. Indeed, there is no universal solution for any problem, but it must be clearly indicated in the text or authors can just avoid making non-justified statements about other solutions.
Author Response
Comments 1: However, there are statements which are simply wrong or just non-objective. Authors placed an opinion (lines 55-57) “However, the limitation of their detection range and the difficulty of detecting 3D magnetic field changes have restricted their wide application in the field of tactile sensing”. I am very sorry to say that without clear definition of the parameter for the detection range this statement is simply not valid. Manuscript still has very narrow citation range for existing fundamental contributions (https://doi.org/10.1007/s10853-015-9643-3;https://doi.org/10.1038/ncomms4266; https://doi.org/10.1088/0957-4484/22/36/365304,etc.).
In addition, proposed solution has strong limitation in sense of the size and temperature range of functionality. Indeed, there is no universal solution for any problem, but it must be clearly indicated in the text or authors can just avoid making non-justified statements about other solutions.
Response 1: Thank you for pointing this out. We fully agree with your suggestions and are very sorry for any inconvenience caused to you due to our lack of rigor. In the manuscript we submitted, we have removed the non-objective descriptive content, and cited the relevant fundamental studies. The modified content has been added respectively from line 53 to line 56 in Section1, and from line 33 to line 36 in Section 1.
“Giant Magnetic Resistance (GMR) [27,28], and Giant Magnetic Impedance (GMI) [29] effects as the sensing elements. These magnetic sensors can detect subtle changes in magnetic fields, enabling high-precision detection of stress signals.”
“Flexible devices exhibit high deformability, generating varying degrees of strain responses when bent, stretched, or compressed. By integrating flexible materials with functional materials through optimized structural design, the limitations of traditional rigid electronic devices are overcome [1-3].”
